# Clinical evaluation of platelet-rich plasma therapy for osteonecrosis of the femoral head: A systematic review and meta-analysis

**Guimei Guo[1,2], Wensi Ouyang[1,2], Guochen Wang[1,2], Wenhai Zhao[1,2], Changwei Zhao[1,2] \***

**1** Changchun University of Chinese Medicine, Changchun, China, **2** Hospital Affiliated to Changchun University of Traditional Chinese Medicine, Changchun, China

\* zcw_1980@126.com

**Data Availability Statement:** All relevant data are within the manuscript and its Supporting Information files.

## Abstract

### Objective

This meta-analysis aims to assess the efficacy and safety of platelet-rich plasma (PRP) for osteonecrosis of the femoral head (ONFH).

### Methods

We comprehensively searched randomized controlled trials in PubMed, Web of Science, EMBASE, the Cochrane Central Register of Controlled Trials, Chinese National Knowledge Infrastructure, China Science and Technology Journal Database, WanFang, and Chinese BioMedical Literature Database from inception until October 25, 2024. The literature on the clinical efficacy of autologous PRP for ONFH was collated. According to the inclusion and exclusion criteria, the literature was screened, quality evaluated and the data was extracted. Meta-analysis was carried out with the software Review Manager 5.4.1 software and Stata 17.0 software. In addition, potential publication bias was detected by the funnel plot test and Egger's test. The GRADE system was used to evaluate the quality of evidence for outcome indicators.

### Results

Fourteen studies involving 909 patients were included in this study. Compared with non-PRP, PRP exhibited significant improvements in the Harris hip score (HHS) at 3 months (MD = 3.58, 95% CI: 1.59 to 5.58, P = 0.0004), 6 months (MD = 6.19, 95% CI: 3.96 to 8.41, P < 0.00001), 12 months (MD = 4.73, 95% CI: 3.24 to 6.22, P < 0.00001), $\geq$ 24 months (MD = 6.83, 95% CI: 2.09 to 11.59, P = 0.0003), and the last follow-up (MD = 6.57, 95% CI: 4.81 to 8.33, P < 0.00001). The PRP also showed improvement in HHS compared to baseline than the non-PRP at 3 months (MD = 3.60, 95% CI: 1.26 to 5.94, P = 0.003), 6 months (MD = 6.17, 95% CI: 3.74 to 8.61, P < 0.00001), 12 months (MD = 5.35, 95% CI: 3.44 to 7.25, P < 0.00001), $\geq$ 24 months (MD = 8.19, 95% CI: 3.76 to 12.62, P = 0.0003), and the last follow-up (MD = 6.94, 95% CI: 5.09 to 8.78, P < 0.00001). The change in visual analog scale (VAS) score 3 months post intervention (MD = -0.33, 95% CI: -0.52 to -0.13, P = 0.001), 6 months

**Funding:** The author(s) received no specific funding for this work.

**Competing interests:** The authors have declared that no competing interests exist.

**Abbreviations:** CI, confidence interval; GRADE, Grades of Recommendation Assessment Development; HHS, Harris hip score; MD, mean difference; MCID, minimally clinical important difference; ONFH, osteonecrosis of the femoral head; PRP, platelet-rich plasma; RCTs, randomized controlled trials; RR, risk ratio; VAS, Visual analog scale.

(MD = -0.69, 95% CI: -0.90 to -0.48, P < 0.00001), 12 months (MD = -0.75, 95% CI: -1.05 to -0.46, P < 0.00001), $\geq$ 24 months (MD = -1.05, 95% CI: -1.20 to -0.89, P < 0.00001), and the last follow-up (MD = -0.75, 95% CI: -0.97 to -0.54, P < 0.00001). The PRP also showed a decrease in VAS score compared to baseline than the non-PRP at 3 months (MD = -0.29, 95% CI: -0.41 to -0.17, P = 0.003), 6 months (MD = -0.63, 95% CI: -0.96 to -0.30, P = 0.0002), 12 months (MD = -0.78, 95% CI: -1.22 to -0.33, P = 0.0006), $\geq$ 24 months (MD = -1.11, 95% CI: -1.27 to -0.96, P < 0.00001), and the last follow-up (MD = -0.74, 95% CI: -1.05 to -0.43, P < 0.00001). Additionally, it was found that the PRP group had the advantages in the following aspects: collapse rate of the femoral head (RR = 0.33, 95% CI: 0.17 to 0.62, P = 0.0006), rate of conversion to total hip arthroplasty (RR = 0.37, 95% CI: 0.18 to 0.74, P = 0.005), and overall complications (RR = 0.33, 95% CI: 0.13 to 0.83, P = 0.02). The GRADE evidence evaluation showed overall complication as very low quality and other indicators as low quality.

## Conclusion

There is limited evidence showing benefit of PRP therapy for treatment of ONFH patients, and most of this evidence is of low quality. Caution should therefore be exercised in interpreting these results. It is recommended that future research involve a greater number of high-quality studies to validate the aforementioned conclusions.

## Systematic review registration

https://www.crd.york.ac.uk/prospero/ #recordDetails, CRD42023463031.

## Introduction

Osteonecrosis of the femoral head (ONFH) is a common refractory disease in orthopedics, characterized by decreased local blood flow and osteoclastic death, which leads to progressive collapse and deformation of the femoral head [1–3]. ONFH patients with hip pain, and intermittent claudication as the main clinical symptoms, if not timely symptomatic treatment, patients gradually develop joint mobility disorders, and in serious cases, affecting normal life and work [4, 5]. The treatment of patients with early to early- and mid-stage ONFH is currently focused on delaying femoral head collapse and protecting hip joint function to improve patient's quality of life. According to the latest guidelines and expert consensus, the currently available standard treatment strategies mainly involve non-weight-bearing therapies, medications, physical therapy, surgical treatments, and other methods, which have some clinical efficacy but still have limitations. Therefore, there is an urgent need to propose new therapeutic options on how to improve the clinical treatment of ONFH [6–8].

It is well known that platelet-rich plasma (PRP) is a platelet concentrate obtained from autologous blood by high-speed centrifugation, which can effectively avoid autoimmune rejection while playing a therapeutic role [9, 10]. PRP contains various types of growth factors, inhibitors of inflammatory factors, and a rich fibrin network. Many of the ingredients synergistically participate in cell proliferation, growth, and differentiation, thereby regulating tissue repair, healing, and regeneration [11–13]. The number of primary studies in this area has increased substantially over the years [14–17]. However, it is inconclusive whether PRP has a therapeutic effect on ONFH. Therefore, this study intends to evaluate the efficacy and safety of

 

PRP in the treatment of ONFH using an evidence-based medicine approach, with a view to providing more therapeutic bases for clinical practitioners.

## Methods and materials

### Protocol register

This systematic review and meta-analysis was structured in adherence to the guidelines of the Cochrane Handbook for Systematic Reviews and was reported as per the Preferred Reporting Items for Systematic Reviews and Meta-Analyses. The protocol was registered in PROSPERO (CRD42023463031) [18, 19].

### Search strategy

Two review authors (G.M.G. and W.S.O.Y.) comprehensively searched four English electronic databases (PubMed, Web of Science, EMBASE, and the Cochrane Central Register of Controlled Trials) and four Chinese electronic databases (Chinese National Knowledge Infrastructure, China Science and Technology Journal Database, WanFang, and Chinese BioMedical Literature Database) from the inception date to October 1, 2023. Search strategies include the keywords below: "platelet-rich plasma", "platelet-rich", "platelet rich plasma", "osteonecrosis of the femoral head", "femur head necrosis", "ONFH", and "FHN". Search approaches included a combination of thematic and free words and were adapted to suit the characteristics of each database. Additionally, the reference lists of included articles were reviewed to obtain as much of the potential research as possible. English search strategy as follows: ((platelet-rich plasma [Title/Abstract]) OR (platelet-rich [Title/Abstract]) OR (platelet rich plasma [Title/Abstract]) AND ("osteonecrosis of the femoral head" [Mesh]) OR ((femur head necrosis [Title/Abstract])) OR (ONFH [Title/Abstract])) OR (FHN [Title/Abstract]). A detailed description of the search strategy used is provided in S1 Table.

### Eligibility criteria

1. Research type: Only published randomized controlled trial studies exploring the clinical outcomes of PRP therapy for ONFH in both Chinese or English languages were considered.

2. Population: Patients who met the diagnostic criteria for ONFH. Patients were not limited to age, gender, ethnicity, and geographical location. Staging references were the Association Research Circulation Osseous stage and the Ficat stage [7, 20].

3. Interventions: The control group received core decompression combined with bone grafting treatment. The treatment group received core decompression combined with bone grafting treatment combined with PRP. PRP treatments are not restricted.

4. Type of outcome measures: Included studies were required to include one of the following outcomes: The primary outcome was the Harris hip score (HHS) (patients were evaluated primarily in terms of pain, function, deformity, and hip range of motion, with a total score of 100. with <70 = poor result, 70–79 = fair result, 80–89 = good result, and >90 = excellent result) [6], Visual analog scale (VAS) score (from 0 to 10, with 0 = no pain and 10 the worst imaginable pain), collapse rate of the femoral head, rate of conversion to total hip arthroplasty, and overall complication.

### Exclusion criteria

1. Literature with overlapping data or duplicate publications.

2. Literature reviews, case reports, animal experiments, basic experimental studies, letters, and review articles.

3. Patients received other treatment during the period of PRP treatment.

4. Primary or relevant outcome indicators are unavailable.

### Data extraction

Two independent review authors (G.C.W. and W.H.Z.) screened the literature for inclusion criteria and exclusion criteria. Any discordance was resolved through in-depth discussions and, if necessary, in collaboration with a third reviewer (C.W.Z.) to reach a unanimous decision. Two independent review authors (W.S.O.Y. and G.M.G.) employed a systematic data extraction template to mine essential study features. The extracted data elements included first authors, publication year, mean age, number of participants and hips, stage of necrosis, type of mechanical support, PRP preparation techniques, PRP application schemes, follow-up time, and outcome indicators. Key outcomes were extracted separately by two other review authors (G.C.W. and C.W.Z.) for data synthesis. A collaborative consensus approach was adopted in data extraction discrepancies between reviewers, involving all reviewers.

### Assessment of the risk of bias

The methodological quality of each included literature was separately assessed by two review authors (G.M.G. and W.S.O.Y.) by using the Cochrane Risk of Bias tool. If the two researchers disagree, it can be resolved by consulting a third reviewer author (C.W.Z.). The quality evaluation includes the following seven aspects: random sequence generation, allocation concealment, blinding of study participants and outcome assessors, completeness of outcome data, selective reporting bias, and other biases [21].

### The Grades of Recommendation Assessment Development and Evaluation

We used the principles of the Grades of Recommendation Assessment Development and Evaluation (GRADE) system to assess the quality of the body of evidence associated with outcomes [22]. Developed to grade the overall certainty of a body of evidence, this approach incorporates five main domains: risk of bias, inconsistency, indirectness, imprecision, and publication bias [23]. Two review authors (G.M.G. and W.S.O.Y.) separately assessed each domain for each selected outcome. If the two researchers disagree, it can be resolved by consulting a third reviewer author (C.W.Z.). We documented all decisions regarding up- or down-grading the certainty of evidence to ensure transparency.

### Statistical analysis

Statistical analysis was conducted using Review Manager 5.4.1 software and Stata 17.0 software. We used the Risk Ratio (RR) for comparisons of binary data and Mean Difference (MD) for comparisons of continuous data. Both measures were accompanied by a 95% Confidence Interval (CI) to encapsulate the effect magnitude when juxtaposing intervention groups and control groups. In each analysis, heterogeneity was tested by the $\chi^2$ test and the $I^2$ value was calculated for quantification. When $I^2$ was less than 50%, it indicated that no significant homogeneity among the included studies, and the fixed-effects model can be used for statistical

analysis. When $I^2$ was greater than 50%, it indicated that the homogeneity among the included studies was obvious, and the random-effects model was used for statistical analysis. To assess the stability of the outcomes, a sensitivity analysis was carried out by excluding the studies one by one at a time. Publication bias was visually assessed by the funnel plot test and Egger's test. We compared pre-to-post-treatment score changes to the minimally clinical important difference (MCID) thresholds determined by previous studies. For HHS, we defined a difference of 10 points to represent the MCID [24]. For VAS score, we defined a difference of 0.9 points to represent the MCID. All selected MCID threshold values were calculated by prior studies via an anchor-based approach [25].

## Results

### Search selection process

During our database search, 448 potentially relevant articles on the treatment of PRP for ONFH were preliminarily retrieved, and 352 duplicate studies were removed by using End-Note software. Then, 61 articles were excluded due to the titles and abstracts that were without high relevance to this study. Two review authors (G.M.G. and W.S.O.Y.) respectively read the full manuscript of 35 publications and excluded 21 of them. Ultimately, 14 published articles [24, 26–38] that met the inclusion criteria were identified for inclusion in this meta-analysis (Fig 1). PRISMA checklist is shown in the S2 Table.

### Characteristics of the included researchs

A total of 909 adult participants with ONFH were included in the 14 eligible studies [24, 26–38]. There were 454 participants in the control group and 455 participants in the treatment group. The detailed characteristics of all the research included are presented in Tables 1 and 2. All studies had clear inclusion and exclusion criteria and there were no significant differences in baseline information between the control groups and treatment groups. The 6 studies [24, 27, 31, 32, 35, 37] were supported by the government or a professional organization. The 8 studies [26, 28–30, 33, 34, 36, 38] did not report the funding. The 9 studies [24, 26–29, 31, 32, 34, 36] followed the Association Research Circulation Osseous classification, and 5 studies [30, 33, 35, 37, 38] followed the Ficat classification. All of the studies described mechanical structural enhancements, 5 of which used β-tricalcium phosphate bioceramic bone [26, 31, 32, 34, 35], 1 of which used porous tantalum rod [27], and 7 of which used autologous bone graft [24, 29, 30, 33, 36–38]. All the studies describe the PRP preparation technique, but the details vary (Table 3).

### Assessment of risk of bias

Among the 14 RCTs, 9 studies [26, 28–33, 35, 38] appropriately described the randomization methods such as random number tables and computerized random methods were rated low risk. Four studies [24, 27, 34, 37] only indicated the use of randomization and did not specify the randomization method, so they were rated as unclear risk. One study [36] did not report randomization methods such as the admission order grouping method was rated high risk. Two studies [30, 35] referred to the use of double-blind methods that were judged at low risk of bias. Three studies [24, 26, 38] conducted blind methods on participants or personnel who were judged at unclear risk of bias. The remaining studies [27–29, 31–34, 36, 37] were judged at high risk of bias. All studies were at low risk for incomplete outcome data. For selective reporting, only two [27, 37] studies reported overall complications and were judged to be low risk. In terms of other biases, All studies were rated as unclear risk (Fig 2).

PRISMA 2020 flow diagram for new systematic reviews which included searches of databases and registers only

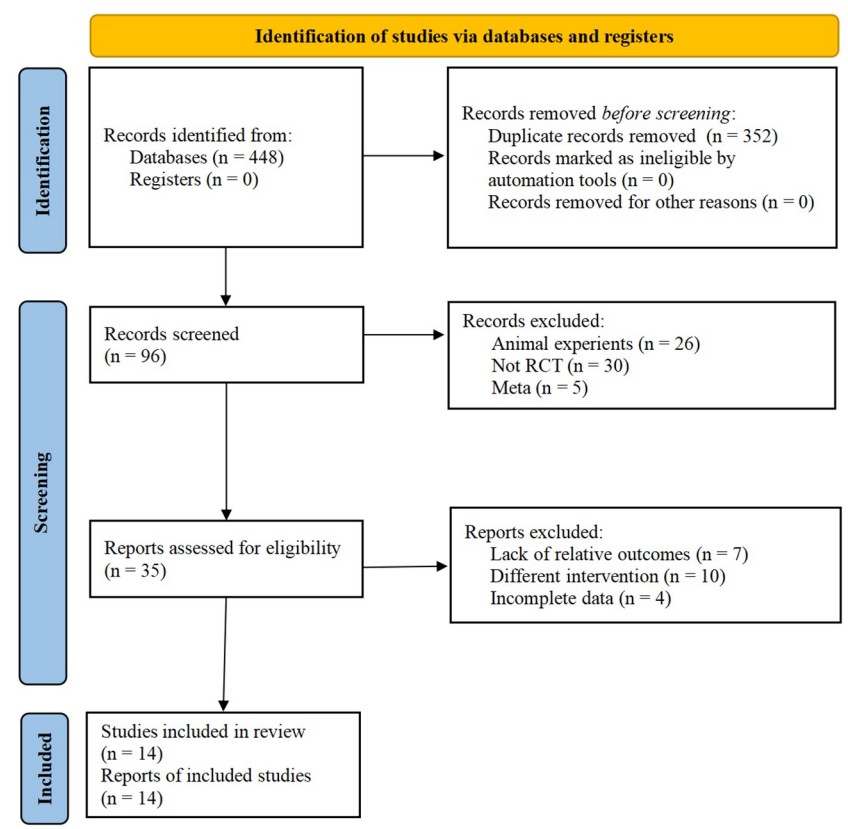

*From:* Page MJ, McKenzie JE, Bossuyt PM, Boutron I, Hoffmann TC, Mulrow CD, et al. The PRISMA 2020 statement: an updated guideline for reporting systematic reviews. BMJ 2021;372:n71. doi: 10.1136/bmj.n71

For more information, visit: http://www.prisma-statement.org/

**Fig 1. PRISMA flow diagram.**

## Meta-analysis results

**Harris hip score (HHS).**   A total of 13 studies [24, 26, 27, 29–38] with 809 participants reported HHS between the treatment groups and control groups. Among the 13 studies, 7 studies [26, 27, 31–35] reported HHS at 3 months postoperatively, 9 studies [26, 27, 29–35] reported HHS at 6 months postoperatively, 9 studies [26, 27, 31, 32, 34–38] reported HHS at 12 months postoperatively, and 3 studies [25, 30, 32] reported HHS at $\geq$ 24 months postoperatively. Meta-analysis was performed on the last follow-up time points, and at different follow-up time points postoperatively (3-, 6-, 12-, and $\geq$ 24-month follow-ups) when those were reported by the respective studies. The meta-analysis revealed overall better treatment group versus control group function although considerable heterogeneity was evident. The results showed that HHS at 3-, 6-, 12-, $\geq$ 24- month, and last follow-ups postoperatively were higher in the treatment group than that in the control group (3 months, MD = 3.58, 95% Cl: 1.59 to 5.58, P = 0.0004; 6 months, MD = 6.19, 95% Cl: 3.96 to 8.41, P < 0.00001; 12 months, MD = 4.73, 95% Cl: 3.24 to 6.22, P < 0.00001; $\geq$ 24 months, MD = 6.83, 95% Cl: 2.09 to 11.56, P = 0.005; last follow-up, MD = 6.57, 95% Cl: 4.81 to 8.33, P < 0.00001). The difference was

**Table 1. Basic characteristics of the eighteen studies included in the meta-analysis.**

| Inclusion studies | Sample (M/F) | Hip (M/F) | Age (years) | Diagnostic standard | Disease stage |
|---|---|---|---|---|---|
| Chai et al., 2022 [26] | T: 29 (13/16)<br>C: 25 (10/15) | T: NA<br>C: NA | T: 43.73 ± 3.25<br>C: 44.33 ± 3.17 | ARCO | II |
| Guo et al., 2022 [27] | T: 60 (28/32)<br>C: 60 (31/29) | T: NA<br>C: NA | T: 50.62 ± 5.25<br>C: 51.12 ± 5.86 | ARCO | II/III |
| Yang 2022 [28] | T: 50 (24/26)<br>C: 50 (22/28) | T: NA<br>C: NA | T: 43.45 ± 7.21<br>C: 43.52 ± 7.19 | ARCO | II |
| Zhang 2021 [29] | T: 41 (29/12)<br>C: 40 (27/13) | T: NA<br>C: NA | T: 37.58 ± 10.26<br>C: 38.72 ± 11.37 | ARCO | II/III |
| Aggarwal et al., 2020 [30] | T: 19 (18/1)<br>C: 21 (19/2) | T: 25 (NA)<br>C: 28 (NA) | T: 38.20 ± 10.40<br>C: 35.20 ± 12.50 | Ficat | I/II |
| Li et al., 2020 [31] | T: 34 (18/16)<br>C: 31 (17/14) | T: NA<br>C: NA | T: 39.17 ± 6.79<br>C: 37.06 ± 7.15 | ARCO | I/II |
| Xian et al., 2020 [24] | T: 24 (9/15)<br>C: 22 (6/16) | T: NA<br>C: NA | T: 28.30 ± 1.40<br>C: 29.60 ± 1.70 | ARCO | II/III |
| Zhang et al., 2020 [32] | T: 52 (48/4)<br>C: 56 (49/7) | T: NA<br>C: NA | T: 41.30 ± 4.70<br>C: 43.10 ± 5.20 | ARCO | I/II |
| Dai et al., 2019 [33] | T: 26 (14/12)<br>C: 26 (15/11) | T: NA<br>C: NA | T: 46.53 ± 1.25<br>C: 46.49 ± 1.21 | Ficat | I/II |
| Wang et al., 2019 [34] | T: 34 (19/15)<br>C: 31 (18/13) | T: NA<br>C: NA | T: 39.29 ± 6.67<br>C: 37.16 ± 7.16 | ARCO | I/II |
| Yuan et al., 2019 [35] | T: 19 (12/7)<br>C: 20 (13/7) | T: NA<br>C: NA | T: 45.00 ± 11.00<br>C: 41.00 ± 14.00 | Ficat | I/II |
| Zhu et al., 2018 [36] | T: 22 (12/10)<br>C: 22 (11/11) | T: NA<br>C: NA | T: 43.23 ± 7.01<br>C: 44.14 ± 5.67 | ARCO | II |
| Zhao et al., 2017 [37] | T: 30 (20/10)<br>C: 30 (19/11) | T: 32 (NA)<br>C: 33 (NA) | T: 40.21 ± 5.10<br>C: 39.25 ± 6.01 | Ficat | I/II/III |
| Yang et al., 2016 [38] | T: 15 (10/5)<br>C: 20 (12/8) | T: 20 (13/7)<br>C: 20 (12/8) | T: 35.60 ± 2.40<br>C: 37.20 ± 7.10 | Ficat | I/II |

ARCO, Association Research Circulation Osseous; C, Control group; NA, Not available; T, Treatment group.

statistically significant but did not reach the set MCID threshold, suggesting that the difference was not clinically significant (Fig 3).

Chang from baseline HHS The results showed that treatment groups were associated with higher HHS from the baseline compared with control groups (3 months, MD = 3.60, 95% Cl: 1.26 to 5.94, P = 0.003; 6 months, MD = 6.17, 95% Cl: 3.74 to 8.61, P < 0.00001; 12 months, MD = 5.35, 95% Cl: 3.44 to 7.25, P < 0.00001; ≥ 24 months, MD = 8.19, 95% Cl: 3.76 to 12.62, P = 0.0003; last follow-up, MD = 6.94, 95% Cl: 5.09 to 8.78, P < 0.00001) (Table 4, S1 Fig).

**Visual analog scale (VAS) score.** A total of 12 studies [24, 26–29, 31–36, 38] with 809 participants reported VAS score between the treatment groups and control groups. Among the 12 studies, 8 studies [26–28, 31–35] reported VAS score at 3 months postoperatively, 9 studies [26–29, 31–35] reported VAS score at 6 months postoperatively, 9 studies [26–28, 31, 32, 34–36, 38] reported VAS score at 12 months postoperatively, and 2 studies [24, 32] reported VAS score at ≥ 24 months postoperatively. Meta-analysis was performed on the last follow-up time points, and at different follow-up time points postoperatively (3-, 6-, 12-, and ≥ 24-month follow-ups) when those were reported by the respective studies. The meta-analysis showed better overall pain relief in the treatment group compared to the control group function although considerable heterogeneity was evident. The results showed that VAS at 3-, 6-, 12-, ≥ 24-month, and last follow-ups postoperatively were lower in the treatment group than that in the control group (3 months, MD = -0.33, 95% Cl: -0.52 to -0.13, P = 0.001; 6 months, MD = -0.69,

**Table 2. Intervention characteristics of included studies.**

| Inclusion studies | Treatment group | Control group | Follow-up (months) | Mechanical support | Outcomes |
|---|---|---|---|---|---|
| Chai et al., 2022 [26] | PRP + CD + BG | CD + BG | 12 | β-tricalcium phosphate bioceramic bone | HHS<br>VAS<br>Femoral head collapse<br>THA |
| Guo et al., 2022 [27] | PRP + CD + BG | CD + BG | 12 | Porous tantalum rod | HHS<br>VAS<br>Overall complication |
| Yang 2022 [28] | PRP + CD + BG | CD + BG | 12 | NA | VAS<br>Femoral head collapse |
| Zhang 2021 [29] | PRP + CD + BG | CD + BG | 6 | Autologous bone graft | HHS<br>VAS |
| Aggarwal et al., 2020 [30] | PRP + CD + BG | CD + BG | 6 | Autologous bone graft | HHS<br>Femoral head collapse<br>THA |
| Li et al., 2020 [31] | PRP + CD + BG | CD + BG | 12 | β-tricalcium phosphate bioceramic bone | HHS<br>VAS<br>Femoral head collapse |
| Xian et al., 2020 [24] | PRP + CD + BG | CD + BG | 36 | Autologous bone graft | HHS<br>VAS<br>Femoral head collapse<br>THA |
| Zhang et al., 2020 [32] | PRP + CD + BG | CD + BG | 24 | β-tricalcium phosphate bioceramic bone | HHS<br>VAS<br>THA |
| Dai et al., 2019 [33] | PRP + CD + BG | CD + BG | 6 | Autologous bone graft | HHS<br>VAS |
| Wang et al., 2019 [34] | PRP + CD + BG | CD + BG | 12 | β-tricalcium phosphate bioceramic bone | HHS<br>VAS<br>Femoral head collapse |
| Yuan et al., 2019 [35] | PRP + CD + BG | CD + BG | 18 | β-tricalcium phosphate bioceramic bone | HHS<br>VAS<br>Femoral head collapse<br>THA |
| Zhu et al., 2018 [36] | PRP + CD + BG | CD + BG | 12 | Autologous bone graft | HHS<br>VAS |
| Zhao et al., 2017 [37] | PRP + CD + BG | CD + BG | 12 | Autologous bone graft | HHS<br>Overall complication |
| Yang et al., 2016 [38] | PRP + CD + BG | CD + BG | 12 | Autologous bone graft | HHS<br>VAS<br>Femoral head collapse<br>THA |

BG, Bone-Grafting; CD, Core Decompression; HHS, Harris hip score; PRP, Platelet-rich plasma; THA, Total hip arthroplasty; VAS, Visual analog scale.

95% Cl: -0.90 to -0.48, P < 0.00001; 12 months, MD = -0.75, 95% Cl: -1.05 to -0.46, P < 0.00001; ≥ 24 months, MD = -1.05, 95% Cl: -1.20 to -0.89, P < 0.00001; last follow-up, MD = -0.75, 95% Cl: -0.97 to -0.54, P < 0.00001), and the differences were all statistically significant. With the exception of ≥ 24-month follow-up results (only two studies included), none of the outcomes met the set MCID threshold, suggesting that the difference was not clinically significant. The cause may occur incidentally and does not represent a true difference in treatment. Therefore, it is uncertain whether PRP treatment has clear clinical significance for pain improvement (Fig 4).

Chang from baseline VAS score The results showed that treatment groups were associated with more lowered VAS score from the baseline compared with control groups (3 months,

**Table 3. PRP preparation techniques and application schemes in the studies.**

| Inclusion studies | Extracted blood volume | Preparation method | Centrifugation parameters | PRP dosage | Application scheme |
|---|---|---|---|---|---|
| Chai et al., 2022 [26] | 30 mL | Centrifugation | 2 centrifugations:<br>• First at 1500 rpm (20 min)<br>• Second at 1500 rpm (10 min) | 3 mL | 1 injection |
| Guo et al., 2022 [27] | 50 mL | Centrifugation | NA | 5 mL | 1 injection |
| Yang 2022 [28] | 30 mL | Centrifugation | 2 centrifugations:<br>• First at 1500 rpm (20 min)<br>• Second at 1500 rpm (10 min) | NA | 1 injection |
| Zhang 2021 [29] | 30 mL | Centrifugation | 2 centrifugations:<br>• First at 1500 rpm (20 min)<br>• Second at 1500 rpm (10 min) | NA | 1 injection |
| Aggarwal et al., 2020 [30] | 50 mL | Centrifugation | 1 centrifugation:<br>• First at 1500 rpm (15 min) | 8 mL | 1 injection |
| Li et al., 2020 [31] | 45 mL | Centrifugation | 2 centrifugations:<br>• First at 2000 rpm (15 min)<br>• Second at 2200 rpm (15 min) | 5 mL | 1 injection |
| Xian et al., 2020 [24] | 90 mL | Centrifugation | 1 centrifugation:<br>• First at 500 rpm (8 min) | 8–10 mL | 1 injection |
| Zhang et al., 2020 [32] | 15 mL | Centrifugation | 2 centrifugations:<br>• First at 1600 rpm (10 min)<br>• Second at 3500 rpm (10 min) | 2.5 mL | 1 injection |
| Dai et al., 2019 [33] | NA | Centrifugation | NA | 30 mL | 3 injections, 3–4 week interval |
| Wang et al., 2019 [34] | 45 mL | Centrifugation | 2 centrifugations:<br>• First at 2000 rpm (10 min)<br>• Second at 2200 rpm (10 min) | 5 mL | 1 injection |
| Yuan et al., 2019 [35] | 48 mL | Centrifugation | 2 centrifugations:<br>• First at 3500 rpm (15 min)<br>• Second at 3500 rpm (20 min) | 5 mL | 1 injection |
| Zhu et al., 2018 [36] | 100 mL | Centrifugation | NA | 4–5 mL | 5 injections, 1 week interval |
| Zhao et al., 2017 [37] | 48 mL | Centrifugation | 2 centrifugations:<br>• First at 2000 rpm (10 min)<br>• Second at 2000 rpm (10 min) | 5 mL | 1 injection |
| Yang et al., 2016 [38] | 48 mL | Centrifugation | 2 centrifugations:<br>• First at 2000 rpm (10 min)<br>• Second at 2000 rpm (10 min) | 5 mL | 1 injection |

MD = -0.29, 95% Cl: -0.41 to -0.17, P < 0.00001; 6 months, MD = -0.63, 95% Cl: -0.96 to -0.30, P = 0.0002; 12 months, MD = -0.78, 95% Cl: -1.22 to -0.33, P = 0.0006; ≥ 24 months, MD = -1.11, 95% Cl: -1.27 to -0.96, P < 0.00001; last follow-up, MD = -0.74, 95% Cl: -1.05 to -0.43, P < 0.00001) (Table 4, S2 Fig).

**Collapse rate of the femoral head.** A total of 7 studies [26, 28, 30, 31, 34, 35, 38] reported the collapse rate of the femoral head between the treatment groups and control groups. Minimal heterogeneity was found (P = 0.83, $I^2$ = 0%), allowing the use of a fixed effect model. The meta-analysis results show better outcomes after PRP therapy (RR = 0.33, 95% Cl: 0.17 to 0.62, P = 0.0006) (Fig 5).

**Rate of conversion to total hip arthroplasty.** A total of 6 studies [24, 26, 30, 32, 35, 38] reported a rate of conversion to total hip arthroplasty between the treatment groups and control groups. Minimal heterogeneity was found (P = 0.93, $I^2$ = 0%), allowing the use of a fixed effect model. The meta-analysis results show better outcomes after PRP therapy (RR = 0.37, 95% Cl: 0.18 to 0.74, P = 0.005) (Fig 6).

**Overall complications.** Only two studies [27, 37] with 180 participants reported overall complications between the groups. In Guo's study [27], one instance of postoperative anemia in the control group and one instance of vein thrombosis in the treatment group were noted.

a.

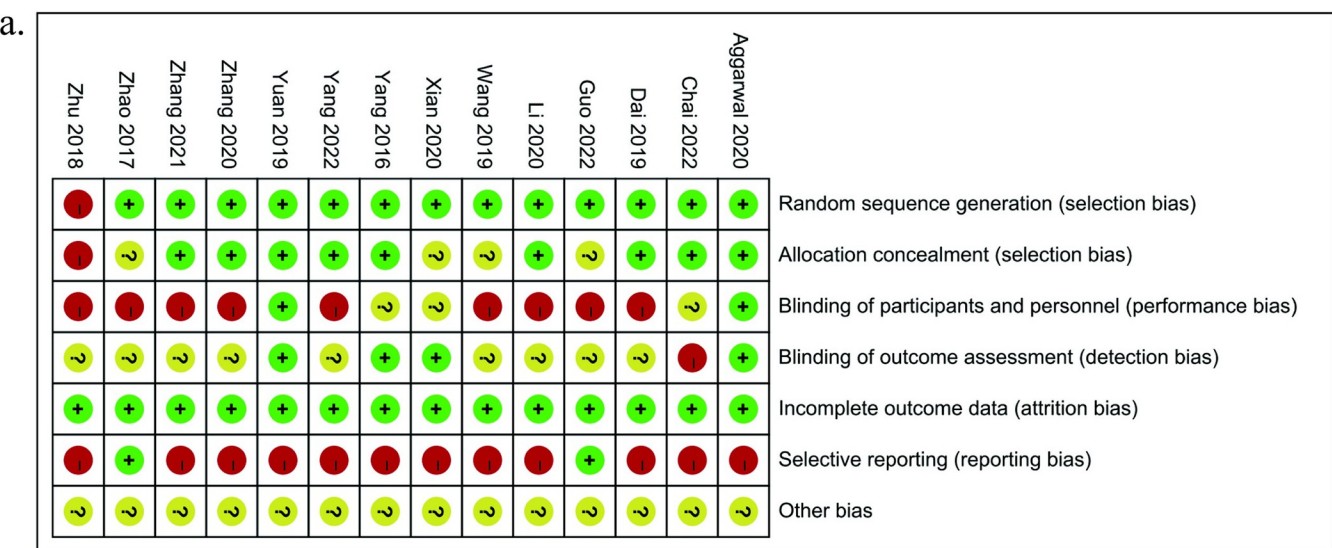

b.

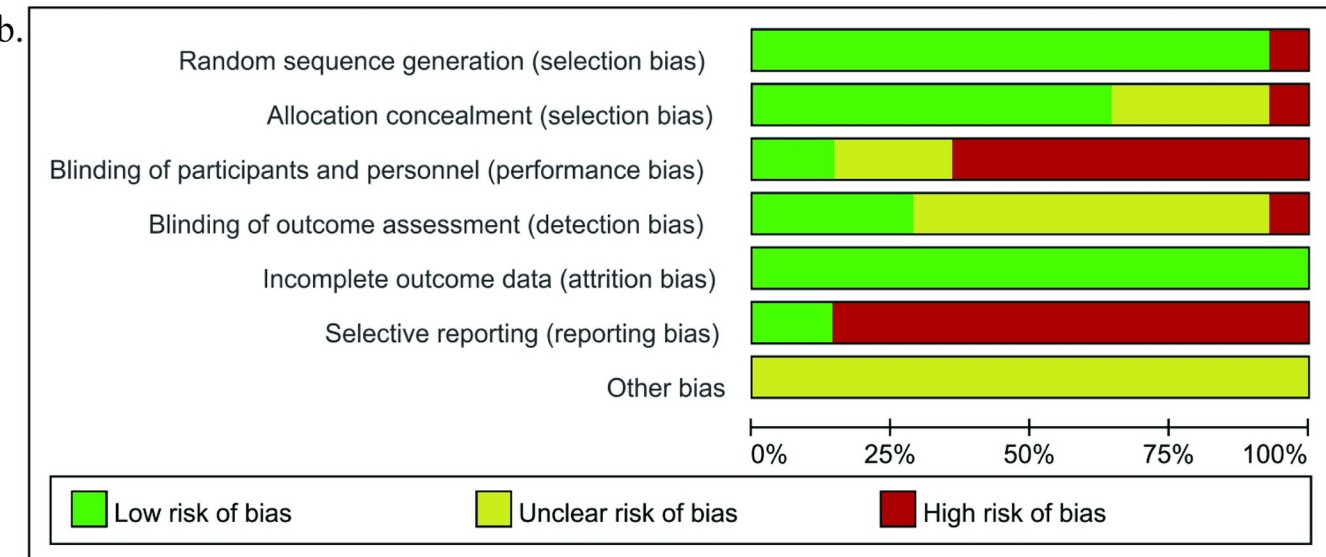

**Fig 2. Risk of bias graph in the included studies: a risk of bias summary b risk of bias graph.**

Zhao's study [37] documented six cases of infection, three instances of hypovolemic shock, four cases of skin redness, and one case of vein thrombosis in the control group and a single case of infection, one instance of hypovolemic shock, and two cases of skin redness in the treatment group. A fixed effect model was utilized given the low heterogeneity (P = 0.40, $I^2$ = 0%). It was statistically significant (RR = 0.33, 95% Cl: 0.13 to 0.83, P = 0.02) (Fig 7). It is worth noting that there is a lack of strict criteria for complications and the result should be treated with caution.

## Sensitivity analysis

We conducted a sensitivity analysis by means of a one-by-one exclusion study, the results of which verified the robustness of our findings. The details of sensitivity analysis are presented in S3 and S4 Tables.

### a. 3-month follow-up

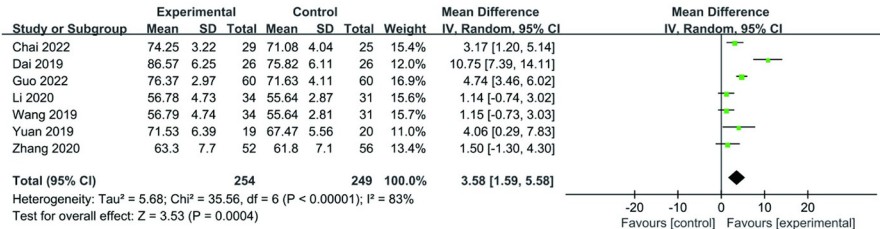

### b. 6-month follow-up

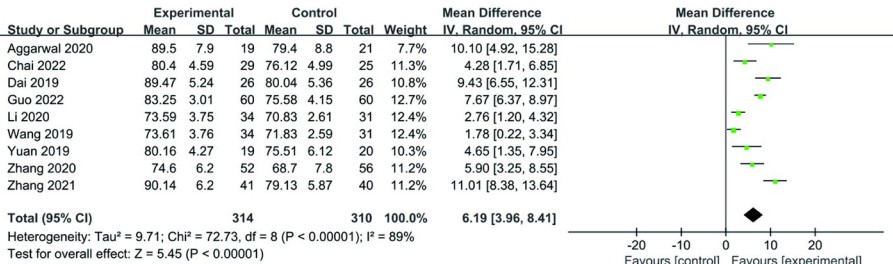

### c. 12-month follow-up

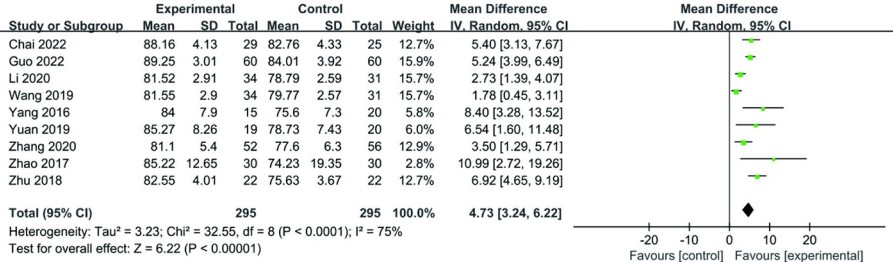

### d. ≥ 24-month follow-up

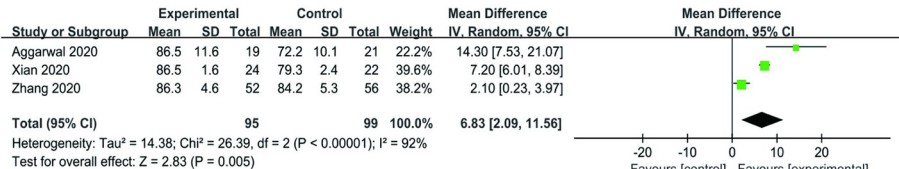

### e. The last follow-up

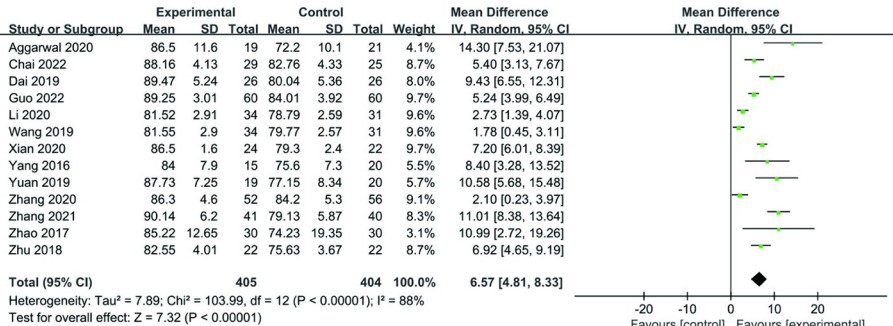

**Fig 3. Forest plot of the meta-analysis comparing the HHS: a the duration of follow-up 3 months b the duration of follow-up 6 months c the duration of follow-up 12 months d the duration of follow-up longer than 24 months e the last follow-up.**

**Table 4. Summary data and analyses.**

| Outcome or subgroup | Statistical methods | Effect estimate | P Value |
|---|---|---|---|
| HHS (values over time)-total | Mean Difference (IV, Random, 95% CI) | 6.57 [4.81 to 8.33] | <0.00001 |
| 3-month follow-up | Mean Difference (IV, Random, 95% CI) | 3.58 [1.59 to 5.58] | 0.0004 |
| 6-month follow-up | Mean Difference (IV, Random, 95% CI) | 6.19 [3.96 to 8.41] | <0.00001 |
| 12-month follow-up | Mean Difference (IV, Random, 95% CI) | 4.73 [3.24 to 6.22] | <0.00001 |
| ≥ 24-month follow-up | Mean Difference (IV, Random, 95% CI) | 6.83 [2.09 to 11.56] | 0.005 |
| HHS (change from baseline)-total | Mean Difference (IV, Random, 95% CI) | 6.94 [5.09 to 8.78] | <0.00001 |
| 3-month follow-up | Mean Difference (IV, Random, 95% CI) | 3.60 [1.26 to 5.94] | 0.003 |
| 6-month follow-up | Mean Difference (IV, Random, 95% CI) | 6.17 [3.74 to 8.61] | <0.00001 |
| 12-month follow-up | Mean Difference (IV, Random, 95% CI) | 5.35 [3.44 to 7.25] | <0.00001 |
| ≥ 24-month follow-up | Mean Difference (IV, Random, 95% CI) | 8.19 [3.76 to 12.62] | 0.0003 |
| VAS score (values over time)-total | Mean Difference (IV, Random, 95% CI) | -0.75 [-0.97 to -0.54] | <0.00001 |
| 3-month follow-up | Mean Difference (IV, Random, 95% CI) | -0.33 [-0.52 to -0.13] | 0.001 |
| 6-month follow-up | Mean Difference (IV, Random, 95% CI) | -0.69 [-0.90 to -0.48] | <0.00001 |
| 12-month follow-up | Mean Difference (IV, Random, 95% CI) | -0.75 [-1.05 to -0.46] | <0.00001 |
| ≥ 24-month follow-up | Mean Difference (IV, Fixed, 95% CI) | -1.05 [-1.2 to -0.89] | <0.00001 |
| VAS score (change from baseline)-total | Mean Difference (IV, Random, 95% CI) | -0.74 [-1.05 to -0.43] | <0.00001 |
| 3-month follow-up | Mean Difference (IV, Fixed, 95% CI) | -0.29 [-0.41 to -0.17] | <0.00001 |
| 6-month follow-up | Mean Difference (IV, Random, 95% CI) | -0.63 [-0.96 to -0.30] | 0.0002 |
| 12-month follow-up | Mean Difference (IV, Random, 95% CI) | -0.78 [-1.22 to -0.33] | 0.0006 |
| ≥ 24-month follow-up | Mean Difference (IV, Fixed, 95% CI) | -1.11 [-1.27 to -0.96] | <0.00001 |

## GRADE evaluation

Based on the principles of the GRADE evaluation, we evaluated the quality of the evidence provided via the HHS, VAS score, collapse rate of the femoral head, rate of conversion to total hip arthroplasty, and overall complication. Table 5 shows that, except for overall complication which was classified as very low-quality, the others were evaluated as low in quality.

## Publication bias

We performed a funnel plot test and Egger's test on the results of ten or more studies. The results showed that no significant publication bias was found in the HHS and VAS score. The details of publication bias are presented in S3, S4 Figs and S5 Table.

## Discussion

It has been a hot topic among doctors to determine the most effective surgical approach for patients with ONFH that will result in the most satisfactory outcome while minimizing the possibility of postoperative complications. Expert consensus on clinical drug prevention and treatment of osteonecrosis of the femoral head (2022), formulated by the Center for Osteonecrosis and Joint-Preserving & Reconstruction, suggests that PRP therapy can induce angiogenesis and osteogenesis, thereby accelerating bone healing and inhibiting inflammatory responses in necrotic lesions [39]. The results of the meta-analysis revealed that PRP therapy could improve the HHS and lower the VAS score, collapse rate of the femoral head, rate of conversion to total hip arthroplasty, and overall complications. Most studies did not meet the MCID threshold for HHS and VAS score, resulting in uncertain clinical relevance. This may have something to do with the fact that the results of our quantitative analysis are highly dependent on MCID values selected from prior studies. These MCID values are highly

### a. 3-month follow-up

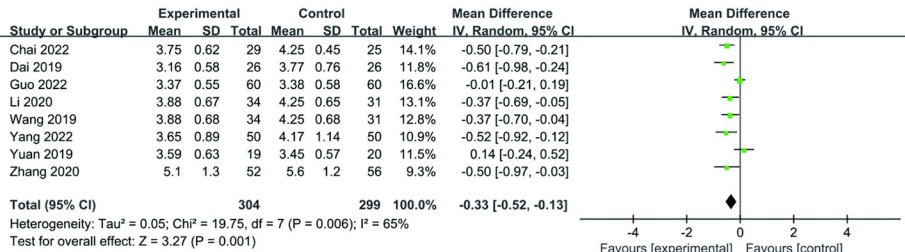

### b. 6-month follow-up

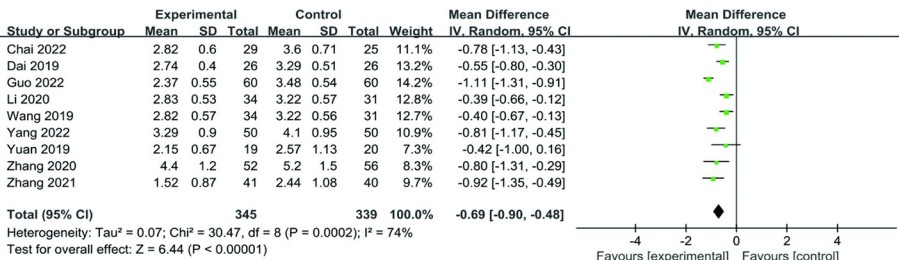

### c. 12-month follow-up

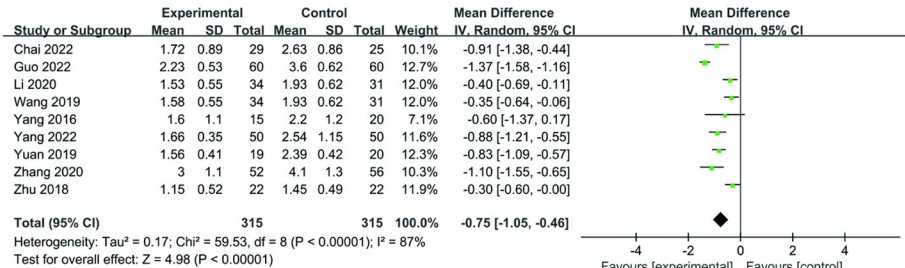

### d. ≥ 24-month follow-up

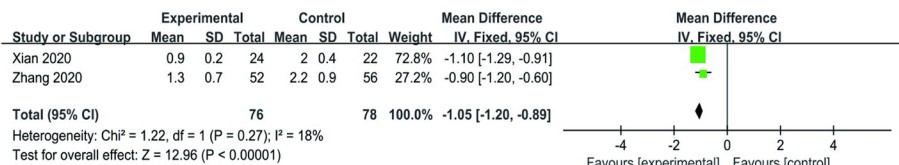

### e. The last follow-up

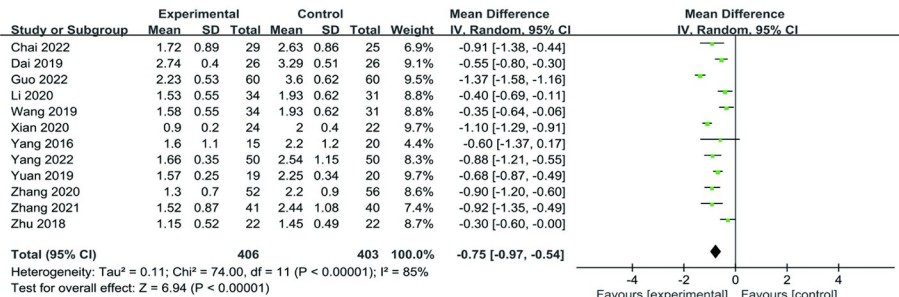

**Fig 4. Forest plot of the meta-analysis comparing the VAS score: a the duration of follow-up 3 months b the duration of follow-up 6 months c the duration of follow-up 12 months d the duration of follow-up longer than 24 months e the last follow-up.**

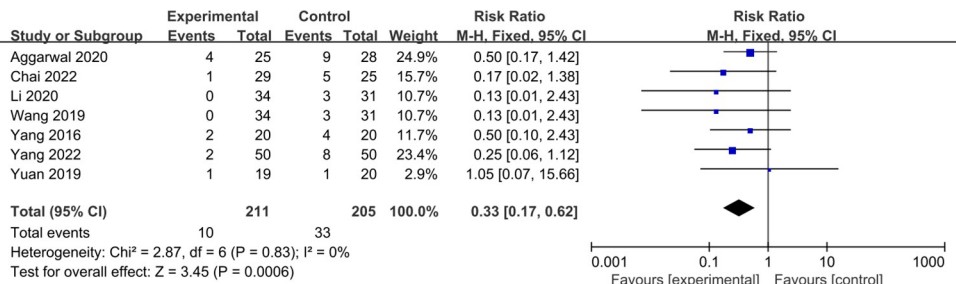

**Fig 5. Forest plot of the meta-analysis comparing the collapse rate of the femoral head.**

impacted by the patient population being studied, and anchor-based approaches are subject to recall bias.

Our review has produced consistent findings with one systematic review published in the English language on PRP for ONFH [40]. Compared to the previous reviews, our review provides a variety of new perspectives. First, more rigorous inclusion criteria and exclusion criteria could increase the quality of evidence. Additionally, the article included the most recent clinical studies that comprehensively assessed important outcome indicators such as HHS, VAS, and femoral head collapse rate, which made this study more convincing and credible.

The HHS has been widely used in the evaluation of clinical efficacy by comprehensively evaluating joint pain, hip function, and daily living abilities. According to a study by Yuan's study [35], HHS at 18 months postoperatively was significantly better in the PRP group than in the non-PRP group, which is consistent with the results of this analysis. Similarly, Chai's study [26] also indicated that the intraoperative use of PRP can reduce patient's pain, improve joint function, and enhance the quality of life, which has some clinical application prospects. Numerous animal experiments have shown that PRP has a positive role in the process of bone repair. Wang's study [41] found in a New Zealand rabbit model that injection of PRP after core decompression promotes the recovery of endosteal trabecular structures in the femoral head. As well, Saginova's study showed that PRP in combination with bone allografts significantly promoted the early stages of bone defect healing [42].

The VAS score is an important indicator for assessing pain in patients. The causes of postoperative pain in patients with ONFH are complex. Intraoperative operations such as scraping of necrotic bone tissue, placement of bone substitutes, and suturing can damage the bone, joint capsule, and blood vessels, resulting in joint pain. Guo's study [27] showed a significant decrease in VAS score for 60 participants who received PRP through intraoperative injections. Surgical trauma and other related injury stimuli can lead to a large release of inflammatory factors such as TNF-α and IL-1β, resulting in the sensitization of peripheral tissues and central

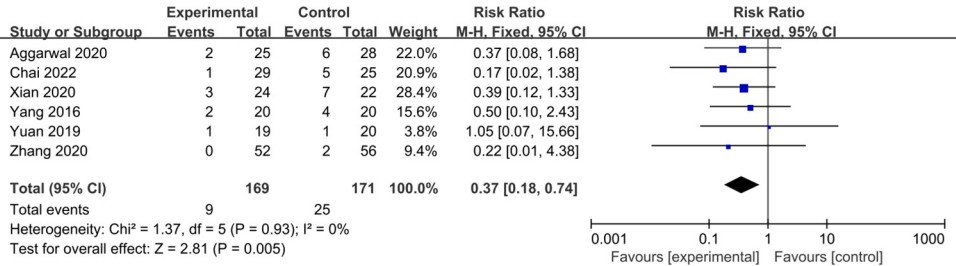

**Fig 6. Forest plot of the meta-analysis comparing the rate of conversion to total hip arthroplasty.**

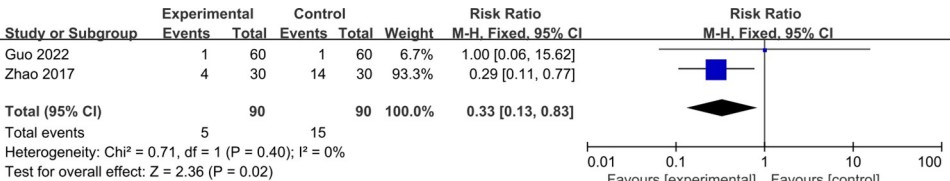

**Fig 7. Forest plot of the meta-analysis comparing the overall complications.**

nervous injury receptors, lowering the patient's own pain threshold, so the body feels pain. Su's study [43] showed that CRP and WBC levels were upregulated to varying degrees in both groups, but they were significantly lower in the PRP group than in the non-PRP group, demonstrating the ability of PRP to improve postoperative analgesic efficacy and downregulate inflammatory factor levels. It is worth noting that scholars have suggested that the pain-relieving effect of PRP may be related to the substance 5-HT in PRP, and a large number of studies have confirmed that 5-HT can influence the transmission of injury to peripheral tissues through receptors and reduce pain levels [44, 45].

For the patient, delaying the collapse of the femoral head is the main principle in the treatment of ONFH. Chen's study indicated that PRP could improve the success rate of core decompression combined with bone graft and effectively reduce the collapse rate of early- and mid-stage ONFH [46]. Similarly, animal experiments by Chen's study demonstrated that PRP promotes the ability of bone marrow mesenchymal stem cells to proliferate and differentiate osteoblasts, and reduces the rate of bone trap vacancies [47]. Some scholar's basic experiments

**Table 5. GRADE evaluation of evidence quality.**

| Outcome | Point estimate (95% CI) | Trials (participants) | Overall quality of evidence | Comments |
|---|---|---|---|---|
| HHS | 6.57 [4.81 to 8.33] | 13 (809) | Low | Risk of bias: All studies at high RoB (downgraded)<br>Inconsistency: $I^2$ = 88% (downgraded)<br>Indirectness: Not downgraded<br>Imprecision: >400 participants<br>Publication bias: No evidence |
| VAS score | -0.75 [-0.97 to -0.54] | 12 (809) | Low | Risk of bias: All studies at high RoB (downgraded)<br>Inconsistency: $I^2$ = 85% (downgraded)<br>Indirectness: Not downgraded<br>Imprecision: >400 participants<br>Publication bias: No evidence |
| collapse rate of the femoral head | 0.33 [0.17 to 0.62] | 7 (416) | Low | Risk of bias: All studies at high RoB (downgraded)<br>Inconsistency: $I^2$ = 0%<br>Indirectness: Not downgraded<br>Imprecision: >300 participants<br>Publication bias: Funnel plot suggests publication bias (downgraded) |
| rate of conversion to total hip arthroplasty | 0.37 [0.18 to 0.74] | 6 (340) | Low | Risk of bias: All studies at high RoB (downgraded)<br>Inconsistency: $I^2$ = 0%<br>Indirectness: Not downgraded<br>Imprecision: >300 participants<br>Publication bias: Funnel plot suggests publication bias (downgraded) |
| overall complication | 0.33 [0.13 to 0.83] | 2 (180) | Very Low | Risk of bias: All studies at high RoB (downgraded)<br>Inconsistency: $I^2$ = 0%<br>Indirectness: Not downgraded<br>Imprecision: Total number of participants <300 events (downgraded)<br>Publication bias: Funnel plot suggests publication bias (downgraded) |

also further confirmed that PRP can promote bone cell growth and bone tissue healing [48, 49].

Meta-analysis results also showed that PRP therapy was effective in reducing the rate of total hip replacement. Zhang's study [32] concluded that the implant's structure facilitates the crawling of neovascularization while mechanically acting as a structural weight-bearing support. PRP therapy through the slow release of active repair factors, promotes neovascular growth crawl through and induces osteoblast proliferation and growth, and then the local presentation of new bone mineralization, accelerates the formation of new bone, and can effectively reduce the number of revision surgeries. Chen's study [46] concluded that PRP may achieve the effect of delaying or even avoiding arthroplasty by controlling infection, inhibiting inflammatory response, and promoting tissue repair, but its long-term efficacy requires further follow-up.

The quality of the RCTs included in our review was assessed by the GRADE system. The GRADE system utilizes a highly structured approach to classify the level of evidence and clearly presents the evaluation items in an item-by-item listing so that clinicians can understand the effectiveness and feasibility of the interventions on their own, and then make clinical decisions. Of the indicators in this study, the overall complications were rated as very low quality, the HHS, VAS score, the collapse rate of the femoral head, and rate of conversion to total hip arthroplasty were rated as low quality, suggesting that there is a discrepancy between the predicted efficacy and the true efficacy of this study. The reasons may be related to the lack of blinding of the included studies, inadequate allocation concealment, large heterogeneity among some studies, and publication bias, which are important issues that need to be addressed in current systematic evaluation studies of the same type. Therefore, it is still necessary to include higher quality RCTs to improve the level of evidence in the future. In the clinical management of patients with ONFH, a comprehensive assessment of the patient's overall condition is still needed to make clinical decisions.

## Limitations and future directions

This systematic review aimed to determine the effectiveness of PRP therapy in managing ONFH. However, there are several limitations to consider: (1) Although the included literature was all RCTs, most articles did not mention randomization methods, allocation concealment, and blinding, which may affect the reliability of the conclusions. (2) There was a high degree of heterogeneity in some of the outcome indicators in the included studies, which may be related to pathogenic factors, diagnostic criteria, and degree of femoral head collapse. (3) Among the included studies, the duration of follow-up was not uniform, and some of the literature had incomplete data information, which may have contributed to some bias in this study. (4) Overall complications have been underreported and safety needs to be further investigated. (5) The lack of standardization of PRP production and protocols for clinical application, makes the PRP products heterogeneous and qualitatively very different from each other, thus limiting the validity of an inter-studies comparison.

To better understand PRP therapy efficacy, addressing these limitations in future research is crucial. With the popularization and application of PRP technology, future research needs to be focused on PRP preparation methods, effective concentrations, intervention doses, and methods of use [50, 51]. In the future, we can focus on patient's coagulation indices, quality of life, and patient satisfaction, making the results of the study more reliable.

## Conclusion

We conclude that there is insufficient evidence to regularly recommend PRP therapy for the treatment of ONFH. In future studies, higher-quality RCTs should be needed to better define PRP therapy as a treatment option for ONFH.

## Supporting information

**S1 Table. Search strategy.**
(PDF)

**S2 Table. PRISMA checklist.**
(PDF)

**S3 Table. Sensitivity analysis for Harris hip score.**
(PDF)

**S4 Table. Sensitivity analysis for visual analog scale score.**
(PDF)

**S5 Table. Publication bias evaluated by egger test.**
(PDF)

**S1 Fig. Forest plot of the meta-analysis comparing the Harris hip score change from baseline: a the duration of follow-up 3 months b the duration of follow-up 6 months c the duration of follow-up 12 months d the duration of follow-up longer than 24 months e the last follow-up.**
(TIF)

**S2 Fig. Forest plot of the meta-analysis comparing the visual analog scale score change from baseline: a the duration of follow-up 3 months b the duration of follow-up 6 months c the duration of follow-up 12 months d the duration of follow-up longer than 24 months e the last follow-up.**
(TIF)

**S3 Fig. The funnel plot of Harris hip score.**
(TIF)

**S4 Fig. The funnel plot of visual analog scale score.**
(TIF)

## Author Contributions

**Conceptualization:** Guimei Guo, Changwei Zhao.

**Data curation:** Guimei Guo, Wensi Ouyang, Guochen Wang, Wenhai Zhao, Changwei Zhao.

**Investigation:** Guimei Guo, Wensi Ouyang, Guochen Wang, Wenhai Zhao.

**Methodology:** Guimei Guo, Wensi Ouyang, Guochen Wang, Wenhai Zhao, Changwei Zhao.

**Software:** Wensi Ouyang.

**Supervision:** Guimei Guo.

**Visualization:** Guochen Wang, Wenhai Zhao.

**Writing – original draft:** Guimei Guo, Wensi Ouyang.

**Writing – review & editing:** Guochen Wang, Wenhai Zhao, Changwei Zhao.

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
