## [Decision Letter · Decision Letter 0]

27 Feb 2024

PONE-D-23-42061Clinical evaluation of platelet-rich plasma therapy for osteonecrosis of the femoral head: a systematic review and meta-analysisPLOS ONE

Dear Dr. Zhao,

Thank you for submitting your manuscript to PLOS ONE. After careful consideration, we feel that it has merit but does not fully meet PLOS ONE’s publication criteria as it currently stands. Therefore, we invite you to submit a revised version of the manuscript that addresses the points raised during the review process.

We look forward to receiving your revised manuscript.

Kind regards,

Sameh Attia, MS

Academic Editor

PLOS ONE

Journal Requirements:

Reviewers' comments:

Reviewer's Responses to Questions

**Comments to the Author**

1. Is the manuscript technically sound, and do the data support the conclusions?

Reviewer #1: No

Reviewer #2: Yes

2. Has the statistical analysis been performed appropriately and rigorously? 

Reviewer #1: Yes

Reviewer #2: Yes

3. Have the authors made all data underlying the findings in their manuscript fully available?

Reviewer #1: No

Reviewer #2: Yes

4. Is the manuscript presented in an intelligible fashion and written in standard English?

Reviewer #1: Yes

Reviewer #2: Yes

5. Review Comments to the Author

Reviewer #1: Platelet-rich plasma has been extensively investigated in the last years in the context of a variety of clinical conditions, including the orthopedic setting. The current study is a systematic review with meta-analysis of platelet-rich plasma therapy for osteonecrosis of the femoral head.

There are several problems with this SR/MA, related to methodological issues, reporting and conclusions, as specified below

ROB ASSESSMENT.

- as shown by the authors, the large majority of studies were at unclear ROB in several domains (namely selection, performance, detection, reporting and also other bias) and this makes the level of available evidence limited or very limited, This need to be clearly stated in the abstract, results and discussion.

OUTCOMES.

- the authors report the Harris hip score (HHS) and VAS score among the outcomes. HHS is a joint specific score that is completed by both the clinician and the patient and consists of 10 items covering domains of pain, function, functional activities, deformity and hip range of motion; VAS is a patient-reported outcome measure. If patients, clinicians, or assessors are aware of treatment assignments, this may influence reporting or measurement of the outcome and introduce bias. Since only 3 studies in the current review referred to the use of blind methods for participants, personnel and assessors, the ROB (performance, detection, reporting) for these 2 outcomes is high. This need to be clearly specified through the manuscript

- Statistical significance and the Minimal Clinical Important Difference (MCID). For bot the outcomes HHS and VAS the authors provide forest plots at different follow-up time, with effect size (MD) and p-value for overall effect. Differences between intervention and controls favour the PRP group in all the comparisons for both outcomes. However, beside the statistical significance (which assess whether the change represents a true treatment difference as opposed to simply occurring by chance) you have to provide information on the clinical importance to patients and clinicians. For instance, the interpretation of outcome using HHS could be as follows: <70 (poor result), 70–79 (fair result), 80–89 (good result) and >90 (excellent result). To this end, you have to provide in the forest plots also the mean values of HHS and related SD at each interval of time, and not only mean difference of the overall analysis with p-values. The mean values need to be presented for both intervention and control, so you can evaluate variations from baseline in both PRP and non-PRP group, together with difference in HHS between the 2 groups useful for the interpretation of MCID.

Similar concerns relate to the outcome VAS: first of all you have to define the MCID for VAS (several examples are available in the orthopedic setting, for example a change of 10 for the 100 mm pain VAS), and then to compare differences between groups and variations from baseline in PRP treated and untreated recipients.

-Outcome “Overall rate of complications”. Not clear why the authors set a priori overall rate of complications among the secondary outcomes and not among primary outcomes. Only 2 trials reported this outcome, and this makes the current SR at high risk of reporting bias for adverse events (the ROB graphs show unclear risk of reporting bias for all the outcomes, but for overall rate of complications should be better to rate it at high risk in 12 of the 14 trials).

In the current SR, the control group received core decompression combined with bone grafting treatment, and the treatment group received the control group combined with PRP. One study has the same rate of complications in PRP recipients and controls, but Zhao’s study [31] documented six cases of infection, three instances of hypovolemic shock, four cases of skin redness, and one case of vein thrombosis in the control group and a single case of infection, one instance of hypovolemic shock, and two cases of skin redness in the treatment group. No information of the type of infection are provided (and not available from PUBMED), and it is really hard to consider PRP responsible of the reduced occurrence of infections compared to controls. A possible explanation is that there was a selection bias between the two groups. The authors need to provide more information related to the Zhao study, particularly on the types of infection

GRADE (Grading of Recommendations, Assessment, Development, and Evaluations) assessment.

The authors have assessed the ROB using the Cochrane but this is not enough for a well conducted SR. You need also to incorporate the GRADE approach to classify the quality of evidence in your review (using a summary of findings table-see https://www.gradeworkinggroup.org/). The GRADE assessment plays a key role in communicating the results of systematic reviews to users. The certainty of a body of evidence involves consideration of within-trial risk of bias (methodological quality), directness of evidence, heterogeneity, precision of effect estimates, and risk of publication bias. I suppose that the level of certainty of the available evidence is very low (or at best low) due to ROB, imprecision and inconsistency (see I2 values for heterogeneity) for most of the outcomes considered, so drawing firm conclusion from these findings it is really problematic.

GRADE assessment should be applied to all the main outcomes of the review.

DISCUSSION AND CONCLUSION: substantial editing and changes are required, using the tools that I have outlined above (e.g., MCID, GRADE, revised ROB) since the level of certainty of the available evidence does not allow to drawn firm conclusions.

Finally, the authors have to discuss the lack of standardisation for PRP production and protocols for clinical application, which makes the PRP products heterogeneous and qualitatively very different from each other, thus limiting the validity of an inter-studies comparison.

Reviewer #2: Dear Editors and Reviewers

Thank you for the opportunity to review the manuscript entitled “Clinical evaluation of platelet-rich plasma therapy for osteonecrosis of the femoral head: a systematic review and meta-analysis”

Summary:

The authors conducted a comprehensive systematic review and meta-analysis of randomized controlled trials to evaluate the clinical effectiveness of PRP in treating patients with ONFH, a topic of significant interest in the field of orthopedics. Their well-written review covers all critical aspects of the topic, including methodological, clinical, and scientific considerations. While the review is commendable, some minor revisions are suggested to enhance its impact.

Comments for Authors:

• Eligibility criteria: The criteria were well-written, but aligning them with the PICOS criteria could enhance clarity. It would be helpful to separate the details of the intervention and comparison groups.

• Outcomes: Providing more detailed descriptions (with relevant literature citations) of each outcome and assessment tool used, such as the Harris Hip Score (HHS), would improve the clarity of the results. Additionally, specifying the overall complication rate would be beneficial.

• Discussing the clinical importance of the effect size is crucial. For instance, in the pain outcome, although the overall effect size at each follow-up duration was statistically significant, it may not have reached clinical significance (>1 point). The same applies to the HHS.

• Including the GRADE system assessment of the certainty of evidence for each outcome would provide an objective basis for future studies.

• Enriching the discussion by comparing the meta-analysis with others in both orthopedic and non-orthopedic fields would further enhance the study's impact. While this is not necessary, it could provide valuable context for readers. Here are some suggested references, though they are not necessary: [https://doi.org/10.1016/j.jogoh.2021.102276, https://doi.org/10.1177/1947603520931170, https://doi.org/10.1155/2022/7487452, https://doi.org/10.1111/coa.13977]

6. PLOS authors have the option to publish the peer review history of their article (what does this mean?). If published, this will include your full peer review and any attached files.

Reviewer #1: **Yes: **Mario Cruciani

Reviewer #2: No

---

## [Author Response · Author response to Decision Letter 0]

28 Apr 2024

Response to the Reviewers

Dear Editor and Reviewers,

Thank you very much for giving us opportunities to revise our manuscript, and we appreciate the reviewer a lot for his positive and constructive comments and suggestions. We have revised the manuscript accordingly, and all amendments are indicated by red font in the revised manuscript. In addition, our point-by-point responses to the comments are listed below this letter. 

We hope that our revised manuscript is now acceptable for publication in your journal and look forward to hearing from you soon. 

With best wishes,

Yours sincerely,

Guimei Guo

First of all, we would like to express our sincere gratitude to the reviewers for their constructive and positive comments.

Replies to Reviewer 

Specific Comments

1. ROB ASSESSMENT.

- as shown by the authors, the large majority of studies were at unclear ROB in several domains (namely selection, performance, detection, reporting and also other bias) and this makes the level of available evidence limited or very limited, This need to be clearly stated in the abstract, results and discussion.

Response: Thank you for your careful review and constructive suggestions regarding our manuscript. In this review, We reworked the ROB analysis. Revised portions are marked in red in the paper (Page 1 Lines 30-31, Page 4 Lines 142-150, Page 17 Lines 477-488). Thanks again for your comment.

2. OUTCOMES.

- the authors report the Harris hip score (HHS) and VAS score among the outcomes. HHS is a joint specific score that is completed by both the clinician and the patient and consists of 10 items covering domains of pain, function, functional activities, deformity and hip range of motion; VAS is a patient-reported outcome measure. If patients, clinicians, or assessors are aware of treatment assignments, this may influence reporting or measurement of the outcome and introduce bias. Since only 3 studies in the current review referred to the use of blind methods for participants, personnel and assessors, the ROB (performance, detection, reporting) for these 2 outcomes is high. This need to be clearly specified through the manuscript.

Response: Thanks for your comments concerning our paper. Those comments are all valuable and very helpful for revising and improving our paper. Revised portions are marked in red in the paper (Page 4 Lines 142-150, Page 17 Lines 477-488). Thanks again for your advice.

3. Statistical significance and the Minimal Clinical Important Difference (MCID). For bot the outcomes HHS and VAS the authors provide forest plots at different follow-up time, with effect size (MD) and p-value for overall effect. Differences between intervention and controls favour the PRP group in all the comparisons for both outcomes. However, beside the statistical significance (which assess whether the change represents a true treatment difference as opposed to simply occurring by chance) you have to provide information on the clinical importance to patients and clinicians. For instance, the interpretation of outcome using HHS could be as follows: <70 (poor result), 70–79 (fair result), 80–89 (good result) and >90 (excellent result). To this end, you have to provide in the forest plots also the mean values of HHS and related SD at each interval of time, and not only mean difference of the overall analysis with p-values. The mean values need to be presented for both intervention and control, so you can evaluate variations from baseline in both PRP and non-PRP group, together with difference in HHS between the 2 groups useful for the interpretation of MCID.

Similar concerns relate to the outcome VAS: first of all you have to define the MCID for VAS (several examples are available in the orthopedic setting, for example a change of 10 for the 100 mm pain VAS), and then to compare differences between groups and variations from baseline in PRP treated and untreated recipients. 

Response: Thank you for reviewing our manuscript and for the constructive comments, which greatly helped us to improve the manuscript. Your Suggestions have helped us to improve the deficiencies in the manuscript and increase the rigor and readability of the article. Revised portions are marked in red in the paper (Page 2 Lines 79-80, Page 3 Line 81, Page 3 Lines 119-122, Page 4 Lines 163-168, Page 5 Lines 180-189, Page 6 Lines 225-228). Thank you very much for your comments.

4. Outcome “Overall rate of complications”. Not clear why the authors set a priori overall rate of complications among the secondary outcomes and not among primary outcomes. Only 2 trials reported this outcome, and this makes the current SR at high risk of reporting bias for adverse events (the ROB graphs show unclear risk of reporting bias for all the outcomes, but for overall rate of complications should be better to rate it at high risk in 12 of the 14 trials).

In the current SR, the control group received core decompression combined with bone grafting treatment, and the treatment group received the control group combined with PRP. One study has the same rate of complications in PRP recipients and controls, but Zhao’s study [31] documented six cases of infection, three instances of hypovolemic shock, four cases of skin redness, and one case of vein thrombosis in the control group and a single case of infection, one instance of hypovolemic shock, and two cases of skin redness in the treatment group. No information of the type of infection are provided (and not available from PUBMED), and it is really hard to consider PRP responsible of the reduced occurrence of infections compared to controls. A possible explanation is that there was a selection bias between the two groups. The authors need to provide more information related to the Zhao study, particularly on the types of infection

Response: Thank you very much for your suggestion. We tried to contact the author of the article, but unfortunately did not receive any response. We are also concerned about this part of the results, so we have added a part of the description. Revised portions are marked in red on the paper (Page 5 Lines 204-205, Page 7 Line 286). Thank you for your marvelous work.

5. GRADE (Grading of Recommendations, Assessment, Development, and Evaluations) assessment.

The authors have assessed the ROB using the Cochrane but this is not enough for a well conducted SR. You need also to incorporate the GRADE approach to classify the quality of evidence in your review (using a summary of findings table-see https://www.gradeworkinggroup.org/). The GRADE assessment plays a key role in communicating the results of systematic reviews to users. The certainty of a body of evidence involves consideration of within-trial risk of bias (methodological quality), directness of evidence, heterogeneity, precision of effect estimates, and risk of publication bias. I suppose that the level of certainty of the available evidence is very low (or at best low) due to ROB, imprecision and inconsistency (see I2 values for heterogeneity) for most of the outcomes considered, so drawing firm conclusion from these findings it is really problematic.

GRADE assessment should be applied to all the main outcomes of the review.

Response: Thank you for reviewing our manuscript and for the constructive comments, which greatly helped us to improve the manuscript. Your Suggestions have helped us to improve the deficiencies in the manuscript and increase the rigor and readability of the article. Revised portions are marked in red on the paper (Page 3 Lines 103-109, Page 5 Lines 210-213, Page 7 Lines 268-278). Thank you for your marvelous work.

6.DISCUSSION AND CONCLUSION: substantial editing and changes are required, using the tools that I have outlined above (e.g., MCID, GRADE, revised ROB) since the level of certainty of the available evidence does not allow to drawn firm conclusions.

Finally, the authors have to discuss the lack of standardisation for PRP production and protocols for clinical application, which makes the PRP products heterogeneous and qualitatively very different from each other, thus limiting the validity of an inter-studies comparison.

Response: Thanks for your comments concerning our paper. Those comments are all valuable and very helpful for revising and improving our paper. Revised portions are marked in red in the paper (Page 7 Lines 287-288, Page 7 Lines 294-295). Thanks again for your advice.

Replies to Reviewer 

Specific Comments

1.Eligibility criteria: The criteria were well-written, but aligning them with the PICOS criteria could enhance clarity. It would be helpful to separate the details of the intervention and comparison groups.

Response: Thank you for reviewing our manuscript and for the constructive comments, which greatly helped us to improve the manuscript. Revised portions are marked in red in the paper (Page 2 Lines 76-78).

2.Outcomes: Providing more detailed descriptions (with relevant literature citations) of each outcome and assessment tool used, such as the Harris Hip Score (HHS), would improve the clarity of the results. Additionally, specifying the overall complication rate would be beneficial.

Response: Thanks for your comments concerning our paper. Those comments are all valuable and very helpful for revising and improving our paper. Revised portions are marked in red in the paper (Page 3 Line 82).

3.Discussing the clinical importance of the effect size is crucial. For instance, in the pain outcome, although the overall effect size at each follow-up duration was statistically significant, it may not have reached clinical significance (>1 point). The same applies to the HHS.

Response: Thanks for your comments concerning our paper. Those comments are all valuable and very helpful for revising and improving our paper. Revised portions are marked in red in the paper (Page 3 Lines 120-123, Page 4 Lines 164-165, Page 5 Lines 181-185, Page 6 Lines 226-229). Thanks again for your advice.

4.Including the GRADE system assessment of the certainty of evidence for each outcome would provide an objective basis for future studies.

Response: Thank you for reviewing our manuscript and for the constructive comments, which greatly helped us to improve the manuscript. Your Suggestions have helped us to improve the deficiencies in the manuscript and increase the rigor and readability of the article. Revised portions are marked in red in the paper (Page 3 Lines 104-110, Page 5 Lines 211-214, Page 7 Lines 269-279).

5.Enriching the discussion by comparing the meta-analysis with others in both orthopedic and non-orthopedic fields would further enhance the study's impact. While this is not necessary, it could provide valuable context for readers.

Response: Thanks for your comments concerning our paper. Those comments are all valuable and very helpful for revising and improving our paper. Revised portions are marked in red in the paper (Page 8 Lines 345-358, Page 9 Line 359, Page 11 Lines 458-461).

We greatly appreciate the efficient, professional, and rapid processing of our paper by your team. If there is anything else we should do, please do not hesitate to let us know. 

Once again, thank you very much for your marvelous comments and suggestions.

---

## [Decision Letter · Decision Letter 1]

7 May 2024

Clinical evaluation of platelet-rich plasma therapy for osteonecrosis of the femoral head: a systematic review and meta-analysis

PONE-D-23-42061R1

Dear Dr. Zhao,

We’re pleased to inform you that your manuscript has been judged scientifically suitable for publication and will be formally accepted for publication once it meets all outstanding technical requirements.

Kind regards,

Sameh Attia, MS

Academic Editor

PLOS ONE

Additional Editor Comments (optional):

Reviewers' comments:

Reviewer's Responses to Questions

**Comments to the Author**

1. If the authors have adequately addressed your comments raised in a previous round of review and you feel that this manuscript is now acceptable for publication, you may indicate that here to bypass the “Comments to the Author” section, enter your conflict of interest statement in the “Confidential to Editor” section, and submit your "Accept" recommendation.

Reviewer #1: All comments have been addressed

Reviewer #2: All comments have been addressed

2. Is the manuscript technically sound, and do the data support the conclusions?

Reviewer #1: Yes

Reviewer #2: Yes

3. Has the statistical analysis been performed appropriately and rigorously? 

Reviewer #1: Yes

Reviewer #2: Yes

4. Have the authors made all data underlying the findings in their manuscript fully available?

Reviewer #1: Yes

Reviewer #2: Yes

5. Is the manuscript presented in an intelligible fashion and written in standard English?

Reviewer #1: Yes

Reviewer #2: Yes

6. Review Comments to the Author

Reviewer #1: all the points raised have been addressed and the quality of the paper has been significantly improved

Reviewer #2: The authors have successfully addressed all my comments. I think this paper is valuable enough for publication. Congratulations.

7. PLOS authors have the option to publish the peer review history of their article (what does this mean?). If published, this will include your full peer review and any attached files.

Reviewer #1: **Yes: **mario cruciani

Reviewer #2: No

---

## [Editor Report · Acceptance letter]

14 May 2024

PONE-D-23-42061R1 

PLOS ONE

Dear Dr. Zhao, 

I'm pleased to inform you that your manuscript has been deemed suitable for publication in PLOS ONE. Congratulations! Your manuscript is now being handed over to our production team.

Kind regards, 

on behalf of

Dr. Sameh Attia 

Academic Editor

PLOS ONE